# Twitter Data Augmentation for Monitoring Public Opinion on COVID-19 Intervention Measures

**Lin Miao[1], Mark Last[1], Marina Litvak[2]**

Ben-Gurion University of the Negev[1], Shamoon College of Engineering[2]
P.O.B. 653 Beer-Sheva 8410501 Israel[1], 56 Bialik St. Be'er Sheva 8410802 Israel[2]
miaol@post.bgu.ac.il, mlast@bgu.ac.il, litvak.marina@gmail.com

## Abstract

The COVID-19 outbreak is an ongoing worldwide pandemic that was announced as a global health crisis in March 2020. Due to the enormous challenges and high stakes of this pandemic, governments have implemented a wide range of policies aimed at containing the spread of the virus and its negative effect on multiple aspects of our life. Public responses to various intervention measures imposed over time can be explored by analyzing the social media. Due to the shortage of available labeled data for this new and evolving domain, we apply data distillation methodology to labeled datasets from related tasks and a very small manually labeled dataset. Our experimental results show that data distillation outperforms other data augmentation methods on our task.

## 1 Introduction

Due to the worldwide social distancing and lockdown restrictions during the COVID-19 pandemic, social media has seen a significant increase in use. Large amounts of publicly available Twitter data can be used for real-time monitoring of the public agreement/satisfaction or disagreement/frustration with intervention measures during the COVID-19 outbreak. This task can be regarded as stance analysis rather than sentiment analysis, because it requires to explore the public opinion on a certain topic rather than just labeling the tweet as literally positive or negative. However, the shortage of labeled Twitter data related to COVID-19 brings a big challenge for the real-time analysis using supervised learning. Therefore, in this work, we focus on the task of automatic data augmentation to allow monitoring the public opinion on COVID-19 measures with a minimal investment in time-consuming manual labeling process.

Our data augmentation approach is inspired by the idea of *data distillation* (Radosavovic et al., 2018). Data distillation is a simple omni-supervised learning method which uses labeled and unlabeled data to update the performance of the model by self-training (Radosavovic et al., 2018; Furlanello et al., 2018; Zhang and Sabuncu, 2020). The previous studies have shown that updated with unseen data in several iterations, data distillation models can reach a consistent improvement.

Our contributions can be summarized as follows:

- We explore several data augmentation methods to deal with the shortage of labeled Twitter data for public opinion monitoring. These methods can be especially helpful when supervised learning methods must be applied in real time.

- We show data distillation to be a feasible solution for the training data augmentation with or without manually labeled data for specific task.

- We create a new labeled dataset of tweets which can be helpful for public opinion analysis on COVID-19 intervention measures.

## 2 Related work

During the COVID-19 pandemic, a lot of research has been done on Twitter data to analyze the public sentiments or responses. To understand the temporal sentiment, (Wang et al., 2020) used lexicon-based model to output sentiment score of each tweet. They also explored the public attitude towards specific measures as a case study. In order to analyze the situation information about the epidemic in social networks, Li et al. (2020) trained a traditional classifier (SVM) on manually labeled data, and applied it to label the rest of the data automatically. In our work, we are interested to explore the public opinion on intervention measures taken by the government during COVID-19

outbreak, which is a different task from sentiment analysis. The public opinion is defined as stance (support, against) on a target topic, regardless of whether positive or negative language is used in the text. Besides, instead of investing in manual labeling of large Twitter datasets, we put efforts on developing an automatic data augmentation system, with or without limited amounts of manually labeled data designed for this particular task.

Data augmentation is critical for supervised learning to tackle the shortage of labeled training data (Dao et al., 2019). To boost the performance of text classification, (Wei and Zou, 2019) proposed to conduct data augmentation using four operations: synonym replacement, random insertion, random swap, and random deletion. Their method requires dataset extension with complicated transformations of the original data. Han et al. (2019) leveraged a large domain-specific corpus to fine-tune a language model for rumor detection, which enabled training data augmentation by weak supervision.

In contrast to these works, we aim to explore automatic data augmentation under weak or no supervision, without the need of data transformations.

The idea of data distillation has been adopted in various semi-supervised learning scenarios on weakly labeled data. Radosavovic et al. (2018) trained a model on large amounts of labeled data to generate the annotations on unlabeled data, then retrained the model using generated annotations. Based on the data distillation framework, the authors of (Liu et al., 2019) distilled predictions from a teacher (complex) model to guide the learning of a simpler student model.

Inspired by the idea of data distillation, we apply it for data augmentation, using two public labeled datasets from related tasks and one small dataset manually labeled by ourselves.

## 3 Methodology

We adopted distillation methods from (Liu et al., 2019) and (Xie et al., 2020) to obtain more labeled data. Unlike these two dealing with image tasks, we adopted distillation method using language model BERT for text analysis task, which is applied to different domain. First, we use a manually labeled dataset to train a basic teacher model. Then, we apply the trained model to unlabeled data to get predicted labels. Following that, we train a student model which is initialized with identical architecture and parameters as the teacher model,

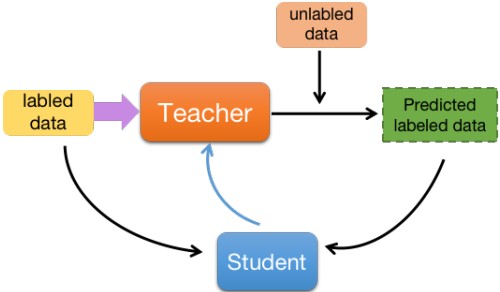

Figure 1: Pipeline of Data Distillation

on the union of manually and automatically labeled data. Later, we use the trained student model as a new teacher model. Then, we iterate the process. During each iteration, only the student model is tested on validation set. Finally, we adopt the model, which provides no further performance improvement on validation set.

The intuition behind this approach is that a high quality teacher model brings up a good student model, and the improvement of the student model will strengthen the teacher model reversely. The better teacher model is, the more accurate labels will be predicted for unlabeled data, leading to better learning for the student model. On the other hand, during each training iteration, the student model evaluates the reliability of the labels predicted by the teacher model to enhance the teacher model. The pipeline of our data augmentation method is shown in Figure 1.

## 4 Datasets

We consider three related datasets for our experiments, obtained from two external sources, which are not from the same task or domain.

**StanceData** is the dataset from SemEval-2016 Task #6 [1] for stance detection in tweets. The dataset is about several different topics, unlike our target task containing only one topic. The training set contains 2224 tweets, in which 1395 are labeled as 'AGAINST', 753 are labeled as 'FAVOR', and 766 are labeled as 'NONE'.

**Sentiment140** is the training data[2] for sentiment analysis, containing 1,600,000 tweets. 800,000 tweets are annotated as positive, and 800,000 tweets are annotated as negative. As opposed to stance analysis, sentiment analysis only concerns about the literal sentiment, though sentiment features are useful for stance detection (Sobhani et al.,

---

[1]http://alt.qcri.org/semeval2016/task6/
[2]http://help.sentiment140.com/for-students

2016). It is noteworthy that this dataset contains only two classes.

**LockdownTweets** is an unlabelled tweet dataset, which is related to the lockdown policy in New York State of United States during COVID-19 pandemic. A large tweet dataset related to the novel coronavirus COVID-19 is provided by (Chen et al., 2020). We conducted our work based on the dataset they contributed. We used Twitter feature 'lang' to select the tweets written in English. Then, we filtered the corpus to obtain only tweets related to New York and lockdown, using a list of keywords. We collected the tweets related to New York using keywords: *New York, NYS, NYC, Governor Cuomo*. Additional keywords were used to select the tweets related to 'lockdown' measure, including: *lockdown, stay-at-home, Pause, NY on Pause, shelter-at-home, NYPause, stay home*.

To monitor agreement or disagreement of public with the 'lockdown' measure, we labeled lockdown-related tweets with 'Support', 'Against', and 'None' labels. We built annotation guidelines (details are shown in Appendix A.1). Based on these guidelines, three annotators reached inter-annotator agreement of Cohen's kappa coefficient equal to 0.872 on 100 samples. Based on these robust guidelines, two annotators manually labeled a larger set of tweets in this collection by reading the content of each tweet. Only tweets assigned the same label by both annotators were used for training and testing. Some representative examples from the manually labeled dataset are shown in Figure 2.

As result, we obtained a labeled dataset containing 475 tweets with 'Against' label, 484 tweets with 'Support' label, and 670 'None' tweets. The remaining 29,394 tweets, dated from 22rd January till 10th June, were left unlabeled. We denote the dataset of unlabeled tweets as **lockdown-unlabel**. The labeled dataset is split into training and test set with the ratio of 2:1, which we call **lockdown-train** (326 Against, 333 Support and 430 None), and **lockdown-test** (149 Against, 151 Support and 240 None).

## 5 Experiments

### 5.1 Experimental settings

**Data preprocessing** The original tweets may contain URLs, which are useless for opinion analysis. So we applied a simple preprocessing of the tweet text, only removing the URLs. All punctua-

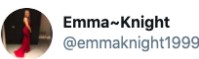

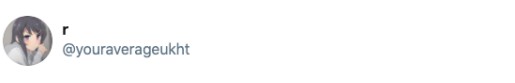

(a) Label: None

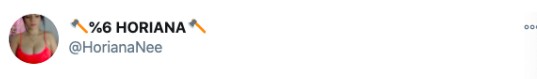

(b) Label: Support

(c) Label: Against

Figure 2: Examples of Manually Labeled Lockdown Tweets

tion, hashtags and special characters were retained.

**Model parameters and training** All experiments are based on fine-tuning BERT (Devlin et al., 2019), in which we could use pre-trained English bert-base-cased model (Devlin et al., 2019) for text sequence representation, which has 12 transformer layers, 12 self-attention heads, and a hidden size of 768. By freezing all the layers of BERT model, we then attached a dense layer and a softmax layer to train the new model. Adam optimizer and Cross entropy loss were used for training all models. Initial learning rate was set to 0.00001, and batch size was set to 32.

To validate the effectiveness of different methods, we conducted experiments on three datasets. Obviously, datasets **StanceData** and **Lockdown-Tweets** show a significant class imbalance. Therefore, we computed class weights for the labels in the training set and then passed these weights to the loss function so that the model will be able to adjust to the class imbalance. Also, when using **Sentiment140** to build models, we output the predicted probabilities, because the models are trained on two-class dataset, but we need to assign three-class labels from the model. To do so, we used predicted probabilities and pre-defined thresholds to assign the label to each instance. Specifically,

| Classifier | P | R | F | Acc |
|---|---|---|---|---|
| *Random* | 0.34 | 0.33 | 0.33 | 0.34 |
| *Majority* | 0.15 | 0.33 | 0.21 | 0.44 |
| *Fully-Supervised* | 0.41 | 0.42 | 0.40 | 0.42 |
| *Transfer-Stance* | 0.35 | 0.35 | 0.32 | 0.41 |
| *Transfer-Sent* | 0.30 | 0.34 | 0.25 | 0.29 |
| Data-Distillation-Self | **0.51** | **0.50** | **0.50** | **0.53** |
| Data-Distillation-Stance | 0.49 | 0.47 | 0.47 | 0.51 |
| Data-Distillation-Sent | 0.37 | 0.39 | 0.32 | 0.34 |

Table 1: Results obtained by different classifiers

when support probability $< 0.4$, the instance is labeled as 'Against', when support probability $> 0.6$, it is labeled as 'Support', otherwise, it is labeled as 'None'.

In all experiments, we evaluated the performance of induced models on **lockdown-test**, using a macro-averaged precision, recall, f1-score, and accuracy.

**Baseline Methods**  We compared data distillation models with the following baselines:
**Random** is a classifier that randomly assigns a label to each given instance.
**Majority** is a classifier that assigns the label to the instance with the majority class.
**Fully-Supervised** is a traditional classifier, by fine-tuning BERT trained on **lockdown-train**.
**Transfer-Stance** is a cross-domain transfer learning (TL) model trained on **StanceData**.
**Transfer-Sent** is a cross-domain TL model trained on **Sentiment140**.
We trained fine-tuned BERT in both TL scenarios, then applied it to the target classification task.

**Data Distillation**  Firstly, we conducted data distillation method on **lockdown-train**. In order to explore the capability of predicting the target labels without any manually labeled data for this particular task, we implemented data distillation method on **StanceData** and **Sentiment140**. In each iteration, 500 new unlabeled tweets from **lockdown-unlabel** were provided for the teacher model to make prediction. Following that, the automatically labeled data was used for training the student model.

## 5.2 Results

Table 1 shows the overall results obtained by different classifiers. It can be seen that the accuracy for the **Majority** baseline is apparently high due to the data imbalance, but the precision and F1 are quite low. It can also be observed that the **Fully-Supervised** classifier has much higher precision and recall than **Random** and **Majority**, but lower accuracy than **Majority**. It is because in the validation

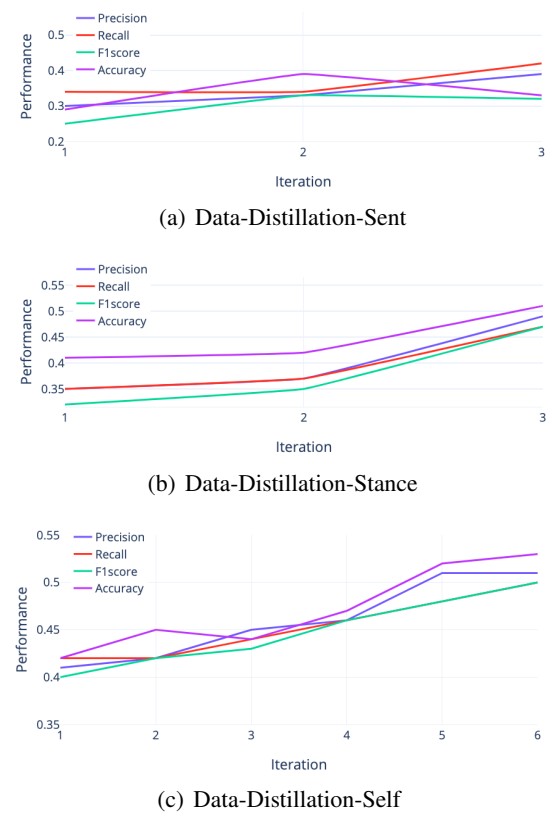

(a) Data-Distillation-Sent

(b) Data-Distillation-Stance

(c) Data-Distillation-Self

Figure 3: Iterative Performance of Distillation Models

set **lockdown-test**, the majority class is much larger than the other two classes, which results in a high accuracy of the **Majority** model.

We can see that **Transfer-Stance** outperforms **Transfer-Sent**. We explain it by the nature of two datasets used for training—while **Transfer-Stance** was designed for the same task (stance classification, but for different topics), **Transfer-Sent** is aimed at sentiment analysis, which is a different task. Besides, **Transfer-Sent** has just two classes, though we heuristically updated it to three classes. In addition, the results of transfer learning are worse than the results of **Fully-Supervised**. It is an expected outcome, when the model trained on datasets imported from other tasks fails to recognize patterns in a different domain. Model **Data-Distillation-Self** obtained the best results with six iterations, and **Data-Distillation-Stance** and **Data-Distillation-Sent** conducted three iterations. Notably, we see obvious improvement from **Data-Distillation-Self**, compared with **Fully-Supervised** model. Figure 3 shows the iterative performance of three distillation models. We could see that iterative learning of the model is effective to improve the performance. Empirically, the student model can consistently improve with the teacher

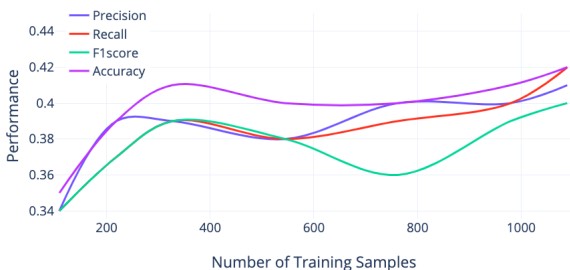

Figure 4: Learning Curve of the Best Teacher Model

model each iteration. After observing more automatically labeled data, the model tends to be more tolerant to the noise, which might be the reason for the improvement. On the other hand, with better initial teacher model (**Fully-Supervised**), **Data-Distillation-Self** outperforms the other two data distillation models. It supports the hypothesis that a higher quality teacher model will finally come up with a better student model. As expected, **Data-Distillation-Stance** and **Data-Distillation-Sent** show significant improvement compared with the transfer learning methods on the same datasets. Because our approach not only transfers a teacher's knowledge to a student, but also adapts it to the domain by providing new instances labeled by a teacher (automatic data augmentation), rather than adopting the pre-trained model in another domain without any further adjustment. However, similarly to the transfer learning, the model built on **StanceData** which is from a more relevant task, tends to have better performance.

Interestingly, we see that **Data-Distillation-Stance** even slightly outperforms **Fully-Supervised**, though it has not seen any manually labeled data. It is a promising indication that our method can achieve data augmentation independently from manually labeled data.

We also explored the performance of the best teacher model method (**Fully-Supervised**) over different amounts of labeled data. The learning curve is presented in Figure 4. We can see an improved performance until all labeled data is available. It indicates that more manually labeled data would lead to better results.

### 5.3 Case Study: Lockdown in NY State

We used the **Data-Distillation-Self** model to automatically assign labels to all unlabeled tweets from **lockdown-unlabel**. Then, we conducted opinion analysis of manually and automatically labeled lockdown related tweets. We assigned 1 for 'Sup-

port', -1 for 'Against', and 0 for 'None'. Then we calculated the daily opinion scores by summing scores from the same date. We also built a policy announcement timeline implemented in New York for COVID-19 ( shown in Appendix A.2). Figure 5 in Appendix A.2 shows the daily analysis aligned with the policy announcement time points. We can observe that, after April, the general public opinion is to oppose the lockdown policy. Before 22th March, when Governor Cuomo announced the statewide stay-at-home order, people did not really care about the lockdown policy. After that, people did not stay indifferent to the lockdown restrictions anymore and the general public opinion had a significant change over time. From the visual alignment of the measures timeline and opinion responses, we can conclude that the measures introduced by government caused a lot of opinionated responses.

## 6  Conclusions and Future Work

In order to monitor public opinion on COVID-19 intervention measures from social media in the future, we adapted data distillation method for training data augmentation. The results on COVID-19 Twitter data show that, using data distillation method outperforms other data augmentation methods. Notably, data distillation method on external sources of data shows bigger improvement over transfer learning on the same datasets. Moreover, the simple but effective data distillation method with a smaller dataset, manually labeled for our specific task, obtains the best performance. In conclusion, we can recommend to invest in manual labeling of a small dataset and further automatically expand it by data distillation for dealing with a specific task with limited annotation resources. In future work, we will apply this method to obtain more labeled data for more comprehensive public opinion analysis on additional intervention measures, like school closure, wearing face masks, etc, and use it for real-time monitoring of public opinion. The maximum required amount of labeled data will also be explored in future work.

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

# A  Appendix

## A.1  Guideline for Annotation

Twitter users express their opinions in different ways, by supporting or opposing the target explicitly, by supporting or opposing some entities related to the target, or by re-tweeting someone's tweet, etc.

**Target**: Lockdown in New York State
**Opinion Labels**: Support, Against, None

**Support/Against**
It can be inferred from a tweet that the user supports/against the target, if:

– The tweet supports/against the target explicitly.
– The tweet supports/against someone or something related to the target, from which we could infer the support/against of the target.
– The tweet does not support or against anything, but it contains some clues can be inferred the support/against of the target.
– The tweet echos the support/against from others.

**None**
None of the two cases above.

– The tweet has neutral opinion on the target.
– Cannot conclude the opinion of the target from the tweet

## A.2  Additional Details for Case Study

Table 2 shows the policies timeline we built according to wikipedia [3].

---

[3]https://en.wikipedia.org/wiki/COVID-19_pandemic_in_New_York_(state)Government_response

| Date | Policy |
|------|--------|
| 7-Mar-20 | State of emergency declared. |
| 9-Mar-2020 | State began producing its own brand of hand sanitizer. |
| 12-Mar-2020 | All gatherings of less than 500 people ordered to cut capacity by 50%. All gatherings of more than 500 people ordered to cancel. |
| 15-Mar-2020 | All New York City schools ordered to close until April 20. |
| 22-Mar-2020 | State-wide stay-at-home order declared. All non-essential businesses ordered to close. All non-essential gatherings canceled/postponed. |
| 25-Mar-2020 | Advisory issued ordering nursing homes to admit patients who test positive for the coronavirus and to not allow testing of prospective nursing home patients. This order was revoked on May 10th. |
| 27-Mar-2020 | All schools statewide ordered to remain closed until April 15. |
| 28-Mar-2020 | All non-essential construction sites ordered to shut down. |
| 6-Apr-2020 | Statewide stay-at-home order and school closures extended to April 29. |
| 9-Apr-2020 | List of businesses deemed essential expanded. |
| 15-Apr-2020 | All state residents ordered to wear face masks/coverings in public places where social distancing is not possible. |
| 16-Apr-2020 | Statewide stay-at-home order and school closures extended to May 15. |
| 1-May-2020 | All schools and universities ordered to remain closed for the remainder of the academic year. |
| 7-May-2020 | Statewide four-phase reopening plan is first announced. |
| 14-May-2020 | State-wide state of emergency extended to June 13. |
| 15-May-2020 | Phase 1 of reopening allowed for counties that met qualifications. Five counties met qualifications and began reopening on this date. Drive-in theaters, landscaping/gardening businesses allowed to reopen state-wide (regardless of Phase 1 qualifications). |
| 23-May-2020 | Gatherings of up to 10 people allowed as long as social distancing is practiced. |
| 8-Jun-2020 | New York City meets conditions for Phase 1, allowing the reopening of construction, manufacturing, agriculture, forestry, fishing, and select retail businesses that can offer curbside pickup. |
| 15-Jun-2020 | Four-phase reopening plan is modified to allow non-essential gatherings of 25 people upon entry of Phase 3, and 50 people upon entry of Phase 4. |
| 22-Jun-2020 | New York City meets conditions for Phase 2, allowing the reopening of outdoor dining at restaurants, hair salons and barber shops, offices, real estate firms, in-store retail, vehicle sales, retail rental, repair services, cleaning services, and commercial building management businesses. |
| 10-Jul-2020 | Malls allowed to open at 25% capacity for regions in Phase 4, with all patrons required to wear masks. |
| 16-Jul-2020 | New restrictions on bars/restaurants only allowing alcohol to be served only to people ordering food. |
| 7-Aug-2020 | Schools allowed to open in-person in the fall if certain conditions are met. |

Table 2: Timeline for Policy Announcement

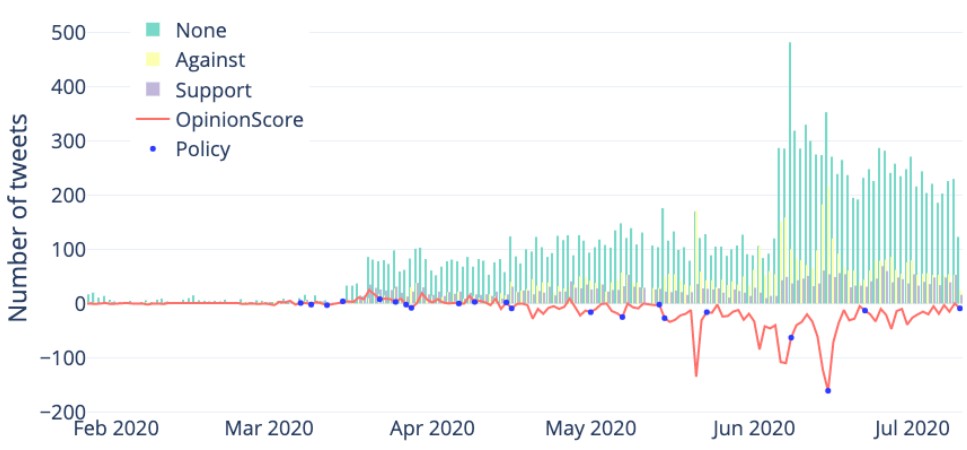

Figure 5: Daily public opinion on lockdown in New York State aligned with the policy announcement timeline