# OpenReview forum: "Twitter Data Augmentation for Monitoring Public Opinion on COVID-19 Intervention Measures"
_EMNLP/2020/Workshop/NLP-COVID — NLP-COVID19-EMNLP Oral_

### Official Review · AnonReviewer2 · 2020-09-16
**Review of Twitter Data Augmentation**

**Rating:** 4
**Confidence:** 3

**Review:**

This study aimed to use Twitter data to better understand stance toward COVID-19 intervention measures. The topic is timely and experiments seemed appropriate. However the approach taken to develop the manual labels was not fully described. In other words, how were the manual labels determined?  It was unclear whether the authors conducted interrater reliability to obtain an understanding of how reliable the manual coding was. Given the models were based on these labeled data, this is rather important to document and demonstrate that the manual data coding process was done in a reliable fashion. To accomplish this the authors need to write out the rules for ’Support’, ’Against’, and ’Neutral’ as well as provide examples (either example tweets or words from the tweets for each case). Without this context it is hard to determine this study ultimate value to the field.

---

> ### Author Response · Authors · 2020-09-27
> **Re: Review of Twitter Data Augmentation**
>
> Thanks for the feedback.
>
> To monitor agreement or disagreement of public with ‘lockdown’ measure, we decided to label lockdown-related tweets on a 3-scale manner, with ’Support’, ’Against’, and ’Neutral’ labels. We collected the lockdown-related tweets for a period of four months , using a list of keywords such as, “New York”, “lockdown”. Then, one annotator manually labeled 1639 tweets in this collection by reading the content of each tweet. The 45 borderline cases were re-assessed by two additional annotators. The final label was set by majority vote for those tweets.
> Due to the unambiguous and simple-to-annotate content for the majority of the collected tweets and the high agreement between the annotators for borderline cases, we believe we got reliable manual labels for the experiments.  We will include more details on the manual annotation process in the revised version of our paper. Below please find some representative examples from the manually labeled dataset.
>
> Example (Against): @NYGovCuomo hysteria? Please lift the lockdown. It’s not necessary and highly detrimental.
>
> Example (Neutral): Since we’re all bored in this #lockdown let’s do an NYSC thread both for the served and serving.https://t.co/l406UaIla5.
>
> Example (Support): i want them to lockdown NY so we can get it over with.

---

### Official Review · AnonReviewer3 · 2020-09-19
**Interesting application of data distillation model on twitter stance analysis**

**Rating:** 6
**Confidence:** 3

**Review:**

The author adopt data distillation model into the tweet stance analysis in support of monitoring public opinion on COVID-19 intervention measures. The research design and experiment results are of good quality.

Some details of the experiments are expected to answer:
* How many rounds for the teacher/student models in data distillation achieves convergence in performance?
* What about the standards and quality control for manual labeling of lockdown-train
(306 Against, 223 Support and 565 Neutral)?
* Figure 2 is not informative sufficiently to support the result.

---

> ### Author Response · Authors · 2020-09-27
> **Re: Interesting application of data distillation model on twitter stance analysis**
>
> Thanks for the feedback.
>
> 1. We mentioned the number of iterations that got the distillation models to achieve convergence in Results section (lines 317 to 320): “Model Data-Distillation-Self obtained the best results with five iterations, and Data-Distillation-Stance and Data-Distillation-Sent conducted three iterations”. We will move this comment near the data distillation description to make it clearer.
>
> 2. Regarding the manual labeling, please see our response to Review 1.
>
> 3. Figure 2 shows the result of our case study, where we tried to visualize public opinion on lockdown measure using automatically labeled tweets with the best data distillation model. However, it was not used to support the results of comparisons between different augmentation methods and cannot demonstrate the accuracy level of automatic labeling using data distillation. The main focus of this paper is data augmentation, as a preparation stage for opinion analysis that will be conducted in our future work.

---

### Official Review · AnonReviewer1 · 2020-09-25
**Good idea but need  more experimental evaluation**

**Rating:** 5
**Confidence:** 4

**Review:**


Authors has used data distillation methodology to augment the data.  The idea is good, and the experiments used seem OK. However, the contribution needs more evaluation. For instant the authors need to plot learning curve to know how Bert model improves with different % of dataset. Second author may investigate how many manually labelled data is required to reach the current performance with data augmentation and see how worthy data augmentation is compared to increase of manually annotated data (how many time and efforts it can save).  Third, I would recommend the authors to investigate the data they have used and check if they are annotated based in robust guideline, it is worth to try this approach in well representative data to see its feasibility. the main issue of some annotated data is that not because it is small but because it is not representative, and they are more vulnerable to noise and overfitting when they are augmented and the performance on test data is likely not a real improvement.

---

> ### Author Response · Authors · 2020-09-27
> **Re: Good idea but need more experimental evaluation**
>
> Thanks for the feedback.
>
> Our research is aimed at monitoring the public opinion on covid-19 intervention measures, which requires large amounts of labeled data. So, in this short paper, we mainly discussed the methods to obtain more labeled data.
>
> Despite investigating BERT improvement over different coverage of a dataset was out of scope for this particular paper, this experiment can be definitely be added to the revised version of the paper.
>
> The second experiment, investigating how many manually labeled data is required to reach the current performance with data augmentation, can definitely add very valuable conclusions, and will be performed in future work, with the extended version of this paper. We thank the reviewer to this suggestion.
>
> Regarding the manual annotation process, please see our response to Review 1. (we hope that the question was understood correctly and the answer is relevant).